# Women and Nature: An Ecofeminist Reading of Chimamanda Ngozi Adichie's *Purple Hibiscus*

**Nigus Michael Gebreyohannes \*** and **Abiye Daniel David**

Department of Foreign Languages and Literature, Addis Ababa University, Addis Ababa NBH1, Ethiopia
\* Correspondence: nigus.michael@gmail.com

**Abstract:** The purpose of this research is to explore ecofeminist issues in Chimamanda Nagozi Adichie's novel *Purple Hibiscus*. It examines the connections between women and nature as well as how unjustified patriarchal domination and Christianity impact these groups as well as indigenous people. A close reading of the novel was conducted in order to select extracts that demonstrate ecofeminist issues. Then, textual analysis was adopted to analyze the selected extracts. Thus, based on the analysis made, the novel shows strong interaction between women and the natural environment. The main character, Kambili, perceives nature as a symbol of hope, freedom, and impressiveness. In contrast, she represents nature as a foreshadowing of chaos and loss of life. The other issue stated in the novel is the women's skill in nurturing plants and flowers. The novel claims that Aunty Ifeoma is knowledgeable and skillful when it comes to gardening. Additionally, Kambili's mother is characterized as an excellent gardener who enjoys caring for the plants and flowers in her garden. Moreover, women are portrayed in the novel as the ones who harvest and produce agricultural goods. Finally, *Purple Hibiscus* illustrates how the patriarchal system and Christianity have led to an unjustified domination of nature and humans based on gender, religion, class, and tradition.

**Keywords:** ecofeminist literary criticism; *Purple Hibiscus*; interaction; domination; nature; women; gender

## 1. Introduction

Post-colonial literature can be considered one of the most important stages of African literature because it addresses various issues in relation to the political, economic, and socio-cultural trends of post-independence countries. It is an important period in African literature because it illustrates how colonialism affected the formerly colonized people. Subsequently, African literature has been reflecting on major African issues. This is because of the emergence of new writers and ever-increasing numbers of women writers, greater awareness of written and oral production in national languages, greater critical attention, and the increase of multiple audiences.

As a result, many authors throughout this time outperformed, and numerous publications on various topics were produced. Thus, bad governance, military coups, corruption, civil war, truth and reconciliation, and migration are among the issues that are common themes in contemporary African literature. However, issues like environmental crises and gender injustice are still serious issues that need further study and discussion. Therefore, in this article, we examine how Chimamanda Ngozi Adichie portrays women and nature in her novel.

One of the great African female writers of this time is Nigerian author Chimamanda Ngozi Adichie, who has earned a great reputation for her novels *Purple Hibiscus* (2003), *Half of a Yellow Sun* (2006), *Americanah* (2013), and her non-fiction work *We Should All Be Feminists* (2014). Adichie is not only a canonic writer but also a good public speaker who has made speeches on several public stages. In line with this, her works have been widely read and studied by scholars from various institutions. Hence, though there is research

and analysis made on her novels, there are still unaddressed issues that can be investigated from various points of view.

Thus, this article explores the ecofeminist issues raised in *Purple Hibiscus* (2003). As a theory, ecofeminism is the combination of ecology and feminism that focuses on the connection and closeness between women and nature and their common unjustified domination and exploitation in a patriarchal society.

It is true that Sagan (2020, p. 138) states, "both as individual animals and as a planetary civilization we [human beings] depend upon the bodies and bodily knowledge of our non-human planet mates." However, ecofeminists believe that, though there is interconnection between human and non-human nature, women and nature have a stronger interdependency than men and nature. The first point that shows that connection is arguably "the concept of maternity: both are mothers"(Valera 2017, p. 12). In addition, Plumwood (1993, pp. 8–9) states that though there are still contradictions in conceptualizing their relationship, it is true that women are connected to nature because of their "empathy, nurturance, cooperativeness, and connectedness reproductive capacity, which are unsharable by men". She adds that the 'backgrounding' and 'instrumentalization' of nature and that of women run closely in parallel.

Similarly, Merchant (1980, p. 75) states that "women and nature have an old-age association, and affiliation that has persisted throughout culture, language, and history". As a result, Warren (2000) identified ten major connections between women and nature.

Likewise, to show the connection between woman and nature in the Indian context, Shiva (1988, p. 37) describes how women are an integral part of nature, both in their imagination and in their daily lives. She states that "Nature is symbolized as the embodiment of the feminine principle on one level, and she is nurtured by the feminine on another level in order to produce life and offer sustenance." In general, since its coinage in 1974, there has been an intensive reflection on the numerous bio-anthropological and socio-cultural connections between women and nature.

Moreover, ecofeminism links nature's exploitation and gender domination and asserts that the origin of this exploitation and domination is patriarchy. Besides, Gaard (1993) states, ecofeminism's basic premise is that the ideology that authorizes oppressions such as those based on race, class, gender, sexuality, physical abilities, and species is the same ideology that sanctions the oppression of nature. Therefore, according to many ecofeminists, "no attempt to liberate women (or any other oppressed group) will be successful until an equal effort is made to liberate nature. As a result, ecofeminists urge the elimination of all forms of oppression" (Gaard 1993, p. 2).

Gaard and Murphy (1998, p. 4) also suggest that ecofeminism is about the interconnectedness of women and nature. It explores "exploitation of nature, the oppression of women, class exploitation, racism, colonialism and Neocolonialism". Furthermore, it opposes and aims to destabilize the hierarchal binary divisions (such as self/other, culture/nature, man/woman, humans/animals, and white/non-white) where men rule as reason, subject, and master, colonizing women as nature, object, and slave.

In this context, Legler (1997) notes that ecofeminism, as a form of literary criticism, is a hybrid form of criticism that combines environmental or ecological critique with feminist literary criticism. It provides literary and cultural critics with a unique blend of literary and philosophical ideas to investigate how nature is represented in literature, as well as how gender, race, class, and sexuality are linked to representations of nature. One of the major goals of ecofeminist literary critics is to investigate the cultural creation of nature, which includes an examination of language, desire, knowledge, and power.

Therefore, this research explores how the aforementioned issues of the interconnectedness and the exploitation of women and nature and other victim groups by a patriarchal system are addressed using the lens of ecofeminism as a literary perspective in *Purple Hibiscus* (2003), which is the primary source of the study. In addition, many secondary sources, like reference books and e-journals, were accessed to obtain further information on the topic. Furthermore, the representation, interdependency, and patriarchal exploitation

and domination of women and nature are used as analytical tools in the examination of the novel. Close reading of the novel was conducted to extract the necessary data, and textual analysis of extracts from the novel was carefully undertaken.

## 2. Results and Discussions

The novel *Purple Hibiscus* is set in Enugu, a city in post-colonial Nigeria, and is narrated by Kambili Achike, a fifteen-year-old girl. *Purple Hibiscus* is a superb novel about adolescent emotional anguish, strong familial relationships, and the bright promise of freedom. By the same token, *Purple Hibiscus* is full of post-colonial issues that are interwoven to make a beautiful story. As a result, it can be examined from numerous angles. However, this study focuses only on the ecofeminist aspects of the novel. Therefore, based on the analysis, the ecofeminist issues raised in the novel have been stated as follows.

### 2.1. Connection between Women and the Natural Environment

The connection between women and nature is one of the issues revealed in *Purple Hibiscus*. As Taylor (2011) states, humans as members of the earth's community have a common relationship with the earth that we share with wild animals and plants. Full awareness of this common relationship gives us a sense of true community with them [members of the earth community]. Thus, Adichie has created an atmosphere in which her female characters interact with the natural environment. Firstly, she has revealed the connection between Kambili, the main character, and nature, which is incorporated as a source of happiness, a symbol of freedom and hope, and, at the same time, a foreshadowing of catastrophe.

Davis (2020) points out that Adichie has established human and non-human interaction. Women are represented as lovers of nature. She also states the role of nature in describing human feelings. This research mainly focuses on the connection between the main characters and nature. However, in addition to these interactions between women and the natural environment, this research focuses on various dominances stated in the novel.

In *Purple Hibiscus*, nature is represented as a source of emotional relief and a symbol of freedom. Kambili, the main character, states that when she saw her father striking her brother, she slowly turned and headed upstairs to change out of her red Sunday dress. Then she sat at her bedroom window after she changed, and the cashew tree was so close she could reach out and pluck a leaf if it were not for the silver-colored crisscross of mosquito netting. She expresses, "The bell-shaped yellow fruits hung lazily, drawing buzzing bees that bumped against my window netting" (Adichie 2003, p. 9). This implies nature's power to heal human emotional distress. After this big turbulence in the house, she was looking outside the house because freedom exists in nature, not in the house where her father caused disruption for the entire family. Thus, rather than her home, it was the natural environment that represented freedom and peace.

Adichie also depicts nature as a source of happiness and grandness. Kambili's sense of connection to the trees, fruits, and flowers in their garden makes her feel fresh and alive. Being entranced by these natural elements, she feels free and pleased. Besides, she feels excited when she smells the wind that brought her the scent of "Sahara and Christmas". She also explains, "Our house still took my breath away; the four-story white majesty of it, with the spurting fountain in front and the coconut trees flanking it on both sides and the orange trees dotting the front yard" to appreciate the beauty of the exterior part of their house. Not only the building, but also the spurting fountain in front, the coconut trees flanking it on both sides, and the orange trees dotting the front yard that give grace to the house and impress her. In this way, she embodied nature as a source of majesty. As stated earlier, Kambili still loves the exterior of their house, which is surrounded by these natural elements. She even watches and listens to the outside world.

Adichie creates a sense of communion between women and nature. Griffin (1994, p. 1) describes how women and nature are intimate, and men consider women to be more a part of nature. She states the following in her prologue:

> He says that woman speaks with nature. That she hears voices from under the
> earth. That wind blows in her ears and trees whisper to her. That the dead sing
> through her mouth and the cries of infants are clear to her. But for him this
> dialogue is over. He says he is not part of this world, that he was set on this world
> as a stranger. He sets himself apart from woman and nature. (Griffin 1994, p. 1)

For this reason, Kambili as a woman interacts with the natural environment actively.
She senses nature by watching, hearing, tasting, and touching. She connects with what
she watches and interacts with what she listens to. That is the reason she states, "… the
rustling of the coconut fronds woke me up. Outside our high gates, I could hear goats
bleating and cocks crowing, and people yelling greetings across mud compound walls"
(Adichie 2003, p. 58). So, this shows Kambili's asset of human life as an integral part of
the natural environment, as Taylor (2011) states. By integrating Kambili with the members
of her natural community, she realized that there was freedom outside the house. Thus,
Adichie has used these elements of nature as a symbol of freedom and hope.

Howell et al. (2013) stated that feelings of connectedness to nature increase one's sense
of "vitality", or "feeling alive", which, in turn, leads to feelings of living meaningfully.
Thus, Adichie states that nature is a symbol of hope and feeling alive. She symbolizes the
rain, sun, freshness of the air, and flowers as hope and continuity of life.

On top of that, women in *Purple Hibiscus* are more closely related to nature than
men. Shiva (1988, p. 73) states that "the backyard of each rural home was a nursery, and
each peasant woman was the sylviculturalist". Adichie has represented women as skillful
gardeners. They constantly plant, weed, cultivate, and inspect their gardens. Kambili
states that the flowers and the trees in their garden were completely the work of her
mother. Her mother is the one who takes care of the flowers, including planting, cultivating,
watering, and decorating them. Besides, Aunt Ifeoma's garden is an indicator that women
are nurtured and have a better sense of connection to the natural environment than men.
To show how her aunt was spending her time in the garden, Kambili states the following:

> Aunty Ifeoma asked me to join them in the garden, to carefully pick out leaves
> that had started to wilt on the croton plants. "Aren't they pretty?" Aunty Ifeoma
> asked. "Look at that, green and pink and yellow on the leaves. Like God playing
> with paint brushes." . . . . Aunt Ifeoma continued watering the row of tiny banana-
> colored flowers that clustered in bunches . . . . (Adichie 2003, pp. 143, 145)

In general, as Plumwood (1993, pp. 8–9) states, the nature of nurturance, cooper-
ativeness, and connectedness to the natural environment is depicted in *Purple Hibiscus*.
Kambili's mother and especially her aunt, Ifeoma, play a significant role in cultivating and
protecting their gardens, which shows women's treatment of nature.

*Purple Hibiscus* also shows the dependency of women on nature. In developing areas of
the world, women are considered the primary users of natural resources (land, forest, and
water) because they are the ones who are responsible for gathering food, fuel, and fodder.
The dependency women have on natural resources, based on their responsibilities, creates
a specific interest that may be different from that of men. In the novel, women are the ones
who harvest and cultivate fruits and vegetables to fulfil their basic needs by consuming
them at home or selling them at the market. However, because of bad governance, war, and
drought, they are vulnerable to a serious problem. The following extract shows a woman
influenced by the devastating war and crisis becoming a victim of poverty. This woman
tries to conceal her hunger at her best, and she states, "Nwunye, things are tough, but we
are not dying yet. I tell you all these things because it is you. With someone else, I would
rub Vaseline on my hungry face until it shone" (Adichie 2003, p. 78).

Not only women, but those indigenous people like Papa Nnukwu's are dependent on
nature, and if nature fails to serve them what they intended, they are exposed to various
problems. Thus, Adichie has stated the impact of poverty on the common people who are
dependent on nature.

On the contrary, in *Purple Hibiscus*, nature has been represented as a foreshadowing
of catastrophe. Haraway (2016, p. 4) states the connection between humans and nature

by saying: "We require each other in unexpected collaborations and combinations, in hot compost piles. We become with each other or not at all." Thus, in *Purple Hibiscus*, Adichie has represented nature as a symbol of hope and freshness and, at the same time, turbulence. Kambili explains that, when her grandfather died, the trees in the front yard bent down, and the flowers were ruffled because the harmattan wind tore across the front yard. Thus, in this case, the harmattan wind is represented as a symbol of destructiveness, instability, and loss of life. Adichie also uses nature as a sign of warning. To express that something bad will happen to her and the whole family, she uses the destructive and dangerous features of nature. She symbolizes nature as a foreshadowing of danger that causes disturbance and instability.

> Everything came tumbling down after Palm Sunday. Howling winds came with an angry rain, uprooting frangipani trees in the front yard. They lay on the lawn, their pink and white flowers grazing the grass, their roots waving lumpy soil in the air. The satellite dish on top of the garage came crashing down, and lounged on the driveway like a visiting alien spaceship. The door of my wardrobe dislodged completely. (Adichie 2003, p. 257)

In this way, Adichie connects humans and nature, especially women and nature. Kambili associates herself with nature. She signifies happiness and freedom with the natural environment. On the other hand, whenever she is unhappy and depressed about circumstances, she symbolizes them with unstable natural phenomena.

### 2.2. Patriarchy Domination of Nature and Women

One of the serious issues described in the novel, *Purple Hibiscus*, is patriarchal domination. (Ling 2014) illustrates that in a core family, the husband gets the opportunity to dominate his wife and his children. Patriarchy is a right hierarchy where the male dominates the female and then extends to all right relationships. In the patriarchal hierarchy, female culture is linked to the body, blood and flesh, material, nature, emotion, and private fields, while male culture focuses on spirit, intelligence, sense, culture, and public fields.

Similarly, in *Purple Hibiscuses*, Papa Eugene, the father, considers himself a rational and dominant man, though he is aggressive and violent. He oppresses and abuses the whole family because he believes that he is the only person responsible for his children and his wife in many ways. Because he is mentally adherent to the colonialists, he acts exactly as the colonialists thought of him, if not worse. He culturally and religiously isolates himself from the indigenous people and tradition.

Ngugi Wa Thiong'o's novel *The River Between*, as Iskarna (2018, p. 191) depicts the relationship between Christianity and colonialism. From a postcolonial standpoint, this novel demonstrates how Christianity is used as an ideological apparatus to construct a myth, doctrine, and perspective in order to force the colonized Kikuyu people to submit to colonial power... Native evangelists and local Christian preachers who have internalized Christianity are used to persuade their people that their traditional religion and customs are associated with the devil. The Bible is interpreted in such a way that native Christians are advised to obey the colonial government. The colonial hermeneutic of the Bible serves as the foundation for the relationship between Christianity and colonialism. The same is true, Seyed and Reyhaneh (2011, p. 53) point out, in Chinua Achebe's Arrow of God; Christianity as an ideological tool gradually changed the minds of African people, leading them to ignore their religion as little more than superstition.

Thus, in *Purple Hibiscus*, religion is used as a reason for domestic violence and a tool to suppress indigenous people and destroy their tradition. Eugene believes that Christianity is the only religion that all the natives should follow. On the other hand, he urges against indigenous beliefs which are matrixed with tradition and rooted in nature. This means, indigenous people consider nature with deep respect, and they have a strong sense of place and belonging. However, he wants them to convert their traditional belief to Christianity.

Therefore, at the beginning, he wants all his family to be fanatic believers and perform all these religious rituals properly though his son Jaja is unwilling to accept everything.

That becomes a reason for Eugene to punishes him mercilessly. Kambili expressed that it was after Palm Sunday that "things started to fall apart at home when my brother, Jaja, did not go to communion and Papa flung his heavy missal across the room and broke the figurines on the etagere" (Adichie 2003, p. 4).

Being a religious man and a domestic abuser, Eugene repeatedly beats his family. One day, Kambili was on her period, and, since she was sick, she had to eat before Mass. Then, after he saw her eating, he was furious. He ignored her pain and viciously attacked his family, which indicates his undermining of femininity and nature. So, Kambili states it as follows:

> He unbuckled his belt slowly. It was a heavy belt made of layers of brown leather with a sedate leather-covered buckle. It landed on Jaja first, across his shoulder. Then Mama raised her hands as it landed on her upper arm, which was covered by the puffy sequined sleeve of her church blouse. I put the bowl down just as the belt landed on my back. (Adichie 2003, p. 103)

As a result, Eugene clearly causes physical and emotional trauma to his family whenever he beats them. Worst of all, he causes several miscarriages for his wife, Beatrice. He brutally beats her and causes several abortions during her pregnancies. This is against femininity, nature, and humanity. He murders his unborn children. This man does not value his family's life; he would rather devote himself to the church. He did this as a result of his colonized mindset.

While this is true, in *Purple Hibiscus*, miscarriage is given a negative connotation by the villagers, which also undervalues femininity. Beatrice was underestimated by the villagers because she had had several miscarriages, though Eugene was the reason. And she states to Kambili what she feels:

> You know after you came and I had the miscarriages, the villagers started to whisper. The members of our umunna even sent people to your father to urge him to have children with someone else. So many people had willing daughters, and many of them were university graduates, too. They might have borne many sons and taken over our home and driven us out, like Mr. Ezendu's second wife did. But your father stayed with me, with us. (Adichie 2003, p. 21)

According to Murphy (2019, p. 10), 'umunna' means "an extended group of paternal kinsmen" in Igbo. As a result, members of these men sent representatives to Eugene to persuade him to have children with other women. Beatrice's miscarriage, according to this group of men, demonstrates her own problem. It reveals that she is no longer a good woman or a good wife. On the other hand, these people ask him to look for other women who are fertile and can bear him many children. In this way, they accept a woman as a fertile human rather than a human, which goes against the natural phenomenon of femininity. Furthermore, their perception proves how they dominate and exploit women to satisfy their egos.

Moreover, Adichie shows Eugene's alienation from his family, roots, and tradition. Eugene believes he is rational, rich, and righteous. He despises his family and traditions. He bitterly hates his father and those who did not convert their religion to Christianity.

This reveals a sense of his disconnectedness from his origin. As a result, he discriminates against and despises those who follow their ancestor's beliefs, referring to them as pagans. Consequently, Kambili claims that her father prayed for "our Papa-Nnukwu's conversion, so that Papa-Nnukwu would be saved from hell. Papa spent some time describing hell, as if God did not know that the flames were eternal and raging and fierce." He did this because he believes that his new religion is the only one that takes someone to heaven. He believes that, because his father did not convert to Christianity, he is a sinner and a pagan, which shows the destruction of indigenous culture. Hence, he advises his children to visit their grandfather but not to eat anything. He tells them, "Go this afternoon to your grandfather's house and greet him. Kevin will take you. Remember, don't touch any food,

don't drink anything. And, as usual, you will stay no longer than fifteen minutes. Fifteen minutes." (Adichie 2003, p. 62)

In contrast, his children try to make a connection with their grandfather though he condemns them not to see him. As a result, one day, Jaja had his grandfather's painting and showed it to Kambili, and she was certain that her father would come and find it. Unfortunately, he came and found it. Then he stanched it from Jaja and did the following:

> The painting was gone. It already represented something lost, something I had never had, would never have. Now even that reminder was gone, and at Papa's feet lay pieces of paper streaked with earth-tone colors. The pieces were very small, very precise. I suddenly and maniacally imagined Papa-Nnukwu's body being cut in pieces that small and stored in a fridge. (Adichie 2003, p. 133)

In this way, Kambili expresses her anger and regret at losing her grandfather's painting. She falls on the ground to collect the pieces of the painting. Next, he ordered her to stand up, but she did not. Then what she remembers is: "The stinging was raw now, even more like bites, because the metal landed on open skin on my side, my back, my legs. Kicking. Kicking. Kicking. Perhaps it was a belt now because the metal buckle seemed too heavy... I closed my eyes and slipped away into quiet." (Adichie 2003, p. 212).

This shows these two kids' sense of connection with their grandfather, tradition, and indigenous identity and how their father disconnected it with a severe punishment. It signifies how obsessed with this colonial mentality and what a victim of it he is. This reflects the impact of colonialism in post-Nigeria. It also reveals the continual colonialism because, at the beginning, it was the missionaries that played a vital role in converting the people. However, by this time, the converted Christians are taking a role in illuminating their ancestral beliefs and traditions. Eugene boldly represents this mentality.

As Khumalo (2019, pp. 26–27) clarifies, Adichie makes a particular reference to Achebe's Things Fall Apart (1958) when she starts her story. As Achebe explores the patriarchal family structure and how it comes undone in the face of colonial European inculturation in the West African region, in *Purple Hibiscus*, Adichie expands the theme and explores the cultural and religious conflict in post-colonial Nigeria.

It is true, Achebe's *Things Fall Apart* addresses the introduction of colonialism and the systematic destruction of tradition. It shows the struggle to resist the new political and religious orders by the white missionaries who were the agents of colonization. Thus, in this way, he uncovers the way the indigenous people struggled against the cultural genocide by the colonizers through his main character, Okonkwo. However, in Adichie's *Purple Hibiscus*, Eugene is the antithesis of this fact. By bearing this colonial mentality, he injects it into his people in the postcolonial period. Thus, Adichie's *Purple Hibiscus* uncovers the genocide of culture led by these natives who perpetuate the colonial legacies.

Moreover, Adichie depicts the domination of the common people because of religious conflict. Therefore, Christianity is revealed as a continuation of colonialism, which discards indigenous beliefs and culture. The natives who converted their religion to Christianity believe that indigenous religions are not true religions. They believe that following such beliefs makes people sinners. As a result, they subdued the natives who persisted in practicing their indigenous religions. Consequently, there are always clashes between these people. So, this is the reason Kambili explains that "Finally, for twenty minutes, Papa prayed for our protection from ungodly people and forces, for Nigeria and the Godless men ruling it, and for us to continue to grow in righteousness" (Adichie 2003, p. 62). He prayed here to be protected from these indigenous people, whom he called "ungodly people and forces" and "Godless men" to the politicians.

Besides, Adichie demonstrates that those who convert their religion are considered superior. Eugene believes, for example, that he is a better person because he is Christian. Since, for him, everything the whites do is right, he believes that being their follower always makes him a right person. As a result, he appreciates his father-in-law because he was liked by the whites since he converted many people's religions. And he says:

Do you know how quickly he learned English? When he became an interpreter, do you know how many converts he helped win? Why, he converted most of Abba himself! He did things the right way, the way the white people did, not what our people do now! (Adichie 2003, p. 67)

In contrast, he hates his father and demonizes him because he did not convert to Christianity. He asked his children, "What did you do there? Did you eat food sacrificed to idols? Did you desecrate your Christian tongue?" (Adichie 2003, p. 70), to ascertain if they had committed any sin by eating anything in their grandfather's home.

The above two extracts show Eugene's perception of his identity and the way he executes his religious beliefs. However, Kambili does not like her father's attitude. She rather appreciates the identity of her paternal grandfather, the one who didn't convert to his religion. This shows her positive view of indigenous culture and identity.

However, the native population, particularly those who did not convert their religion, undervalues individuals who became Christians. They despise and ridicule them because they think they adhere to the beliefs and customs of white people. One of these men tells Eugene, "You are like a fly blindly following a corpse into the grave!" to show that his conversion of religion had no apparent reason based on their understanding.

Adichie has also reflected on powermongers and the destruction they cause to ordinary society. She explores how keen men are crazy about seeking power and how they exploit people. Military men blindly kill each other and exploit women. Politicians are also corrupt and have failed to serve the people effectively. They take people's money and make it their own. That is evidence of how men exploit innocent people, including women and children, in the novel. Here is the excerpt:

A coup always began a vicious cycle. Military men would always overthrow one another, because they could, because they were all power drunk. Of course, Papa told us, the politicians were corrupt, and I the Standard had written many stories about the cabinet ministers who stashed money in foreign bank accounts, money meant for paying teachers' salaries and building roads. (Adichie 2003, p. 25)

*Purple Hibiscus* also reveals the negative consequence of women's dependency on men. There might be reasons that make women dependent on men. As a result, they lose their courage after something happens to their husband, like death or divorce, because they believe that they cannot raise their children alone. They may be vulnerable to many problems. A woman says, "What will I do, sir? I have three children! One is still sucking my breast! How will I raise them alone?" (Adichie 2003, p. 38) after the death of her husband, who was responsible for the whole family.

In addition, the novel, *Purple Hibiscus,* shows the exploitation and operation of women and nature during wartime. Men, mostly military members, rape and kill women with no sympathy. They torture innocent people for the sake of realizing their power. This vividly shows males' domination of women. The extract also shows that women are naturally dependent on nature. They grow vegetables and feed themselves. This shows how women's dependency on nature and the elements is subjugated to men's aggressiveness. As in the novel, women are innocent; they cultivate vegetables and sell them to the market while men kill and damage society.

In the outskirts of the market, we let our eyes dwell on the half-naked mad people near the rubbish dumps, on the men who casually stopped to unzip their trousers and urinate at corners, on the women who seemed to be haggling loudly with mounds of green vegetables until the head of the trader peeked out from behind. Market women were shouting, and many had both hands placed on their heads, in the way that people do to show despair or shock. A woman lay in the dirt, wailing, tearing at her short afro. Her wrapper had come undone and her white underwear showed. (Adichie 2003, p. 45)

## 3. Conclusions

In *Purple Hibiscus*, Adichie shows the connection between the human and non-human environment. She has reflected her ethic on the earth and natural environment. The connections show that humans are one member of the earth's community. On top of this, she depicts women as more connected and closer to the natural environment, which makes the novel open to being analyzed from an ecofeminist literary perspective.

Therefore, based on the analysis made, the novel has major ecofeminist issues stated in many ways. The first thing is that it shows the interaction between women and nature. In this case, the protagonist, Kambili, portrays her strong connection to the natural phenomenon. She senses these parts of nature and expresses what she feels in many ways.

For this reason, Kambili portrays the healing power of nature. Besides, she represented nature as a symbol of hope, love, and freedom. She gives meaning to what she listens to, watches, and touches, and creates a sense of communication with the natural environment. Whenever she is sad and anxious, she tries to rediscover her happiness in the natural environment. She loves to watch nature outside her house because the external world brings her relief, freshness, hope and grandness. On the other hand, Kambili shows that nature symbolizes turbulence and instability. The destructive side of nature represents an overshadowing of bad incidents in her and her family's life. In this manner, Adichie shows the sense of communion between women and nature extensively.

Furthermore, in *Purple Hibiscus*, not only Kambili but her aunt Ifeoma and her mother are also close to and interlinked with the natural environment. They are represented as nurturers of plants and flowers in their backyard. They do this because it gives them pleasure and emotional relief. Thus, this reveals their closeness and independency to the natural environment. In addition, by managing their gardens, they can still contribute to reducing environmental pollution in cities. Thus, the narrator has demonstrated that women still have a propensity to preserve the environment, whether they reside in urban or rural locations.

The novel also emphasizes women's reliance on nature. Women in third-world countries are dependent on nature. Their day-to-day activities are cultivating, farming, weeding, and harvesting. Their way of living is simply based on what they do in relation to nature. As a result, the women in the novel practice all these activities. Thus, one can argue that women are dependent on nature, and it is their means of living, in short.

The novel also depicts the uprooting of culture, tradition, and religion, as well as an alienation from nature. Eugene is a devout Christian. Being a Christian means being rational, civilized, and righteous. As a result, he expects everyone, especially his family, to think and act the way he does. So, if his family deviates from his rule, he brutally abuses them and causes them physical and psychological harm. He even disregards his daughter's and wife's femininity, demonstrating his patriarchy and undervaluation of nature. Furthermore, he encourages others to abandon their traditional and indigenous beliefs in order to convert to Christianity. Despite the fact that indigenous beliefs are deeply rooted in nature and that the native people accept and value nature as an integral part of their existence, he pushes the people to convert their religion. However, his children respect their tradition and their grandfather, who stands in for their true identity and origin, regardless of the fact that there is perpetual conflict between him and those who don't listen to and follow him, including his father.

At last, the novel signifies that women and nature are highly dominated and exploited by patriarchal power. They are victims of the atrocities of war and conflict. In this way, Adichie uncovers the religious, traditional, and social conflicts in postmodern Nigeria.

**Author Contributions:** N.M.G. conceived the idea of the research and wrote the manuscript under the supervision of A.D.D. All authors have read and agreed to the published version of the manuscript.

**Funding:** This research received no external funding.

**Acknowledgments:** We want to thank our families and friends who helped and inspired us while we were conducting this research.

**Conflicts of Interest:** The authors declare no conflict of interest.

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
