# Peer review of "Women and Nature: An Ecofeminist Reading of Chimamanda Ngozi Adichie’s Purple Hibiscus"

_2410-9789, doi:10.3390/literature2030015_

Round 1

Reviewer 1 Report

This article displays some insightful and interesting points on how Adichie's novel depicts, and engages with, ecocritical issues. The link between women and nature is overall clearly present in the novel, and is rightly given prominence in the article.
However, as it stands, the article is more a summary of examples where women engage in nature and how the patriarchy either oppresses them, or counters this relation with nature (eg through the Christian religion). No engagement with secondary sources is thoroughly expanded on: while the introduction presents an overview of existing ecocritical theories and key terms, they never return in the main body of the text. The conflation nature-mother is somewhat simplistic and, on page 2, the authors' discussion of Gaard (and how it is the patriarchy that has pigeon-holed women into the 'natural domain') seems to contradict their previous position about women being naturally more linked to nature. Clarification of this point is needed.
The lengthy quotes provide good examples that the text has been understood, but the authors need to provide an overall research question and argument to pursue throughout the essay, in order for it to be an innovative contribution to scholarship.
This reviewer recommends the following changes:
- consider cutting the methodology section by integrating it in the introduction. This will allow a clearer engagement with the secondary sources and constructing a clearer argument.
- restructuring the essay around the themes, instead of around the types of examples. Reduce the number and lenght of quotes; perform close readings eg of recurring figures of speech. Use secondary sources discussing both Adichie's work and the ecocritical questions at stake throughout, in order to produce a more original, thorough analysis.
- the relationship between Christianity, the past (and present) role of Christian missionaries in Nigeria and the patriarchy needs to be clarified further. Some background reading into how missionaries cemented patriarchal systems and led to quasi-genocidal cultural assimilation, at least via Achebe's Things Fall Apart, is needed.

- engage with the following secondary sources:

Donna Haraway, Staying with the trouble (2016)

Val Plumwood, Feminism and the mastery of nature (1993)

Sincy Davis, Narrating Human Ecology in Adichie's Purple Hibiscus (RJELAL, 2020)

Sibongile Khumalo, Under the Hibiscus: An eco-critical reading of Chimamanda Ngozi Adichie's Postcolonial Novels (MA thesis, 2019)

In terms of style and language:
- proofread attentively (eg Adichie's name is misspelled in the abstract and elsewhere as 'Adiche'; p. 2: 'in to' should be 'into')
- choose a tense: either present or past and stick to it throughout.

Reviewer 2 Report

The idea of an ecofeminist reading of the novel is a good one.  The author does in fact make insightful points and highlight key passages in support of the reading.

However, the article is poorly written.  Its organization is reminiscent of both a literature review and the methodology section of a dissertation/thesis.  Much of what is included is unnecessary and superfluous.

It is also clear from the diction and syntax that English may not be the author’s first language.  This can be remedied by employing an editor.

Author Response

Please see the attachemnet.

Round 2

Reviewer 1 Report

Thank you for sending the revised version, which shows great improvement with respect to the first manuscript. You have read an impressive amount of secondary (and primary) sources over this short period and I'm glad my suggestions are useful.

Some relatively minor changes would still enhance the cohesion and coherence of the argument: as it stands, your position still seems to be that Adichie represents a relation between women and nature, and that Christianity is a legacy of colonialism. How the two are connected, and how this is an original argument needs to be emphasised, no less in the abstract, which (as in the previous version) summarises the novel and does not fully address the aims and results of the article.
I suggest making the following changes:
- redraft the abstract, clarifying research aims, methods (in brief), and research question.
- integrate quotes from primary and secondary sources more organically within argument. Eg quote by Griffin on p.3 (can it be presented instead on p. 4, where you actually discuss it?)
- pp.4-5: the grandfather's death is mentioned almost in passing on p.4 and then, on the next page, is discussed more in detail, in relation to how Indigenous people live and experience nature. Can the two be combined and reduced to one, stronger close reading like the one of the Sunday dress and the tree scene?

- p.6: comparison to Fulani women is weak and does not provide evidence from text. Consider omitting it and replacing it with a sustained comparison between Kambili and her period, with her mother and the miscarriages (and the role of the father in both instances)

- p. 7: paragraphs on Achebe's Things Fall Apart (which I am pleased you found useful!) need to be linked better to the question of giving birth, which comes in the subsequent paragraph. Unless of course you change the structure completely, as I suggest above.

- p.8: paragraph on Christianity repeats what has been said earlier.

- p. 9: the first two paragraphs (before the conclusion) can be omitted.
Some suggestions on form:
- proofread carefully and check references: there are several instances where no page numbers are given for quotes from primary and secondary sources, and/or where punctuation spacings are not standardised.

- try to avoid repetitions, eg. phrases such as 'as has been stated before'. if you need to engage with the same example or secondary source again, you may quote a different sentence from the same book/article, and discuss it in greater depth.
